# Nogo-A Modulates the Synaptic Excitation of Hippocampal Neurons in a Ca^2+^-Dependent Manner

**DOI:** 10.3390/cells10092299

**Published:** 2021-09-03

**Authors:** Kristin Metzdorf, Steffen Fricke, Maria Teresa Balia, Martin Korte, Marta Zagrebelsky

**Affiliations:** 1Division of Cellular Neurobiology, Zoological Institute, TU Braunschweig, D-38106 Braunschweig, Germany; k.metzdorf@tu-bs.de (K.M.); m.balia@tu-bs.de (M.T.B.); m.korte@tu-bs.de (M.K.); 2Helmholtz Centre for Infection Research, AG NIND, Inhoffenstr. 7, D-38124 Braunschweig, Germany; 3Division of Cell Physiology, Zoological Institute, TU Braunschweig, D-38106 Braunschweig, Germany; steffen.fricke@tu-braunschweig.de

**Keywords:** Nogo-A, excitation/inhibition (E/I) balance, calcium-permeable (CP-) AMPARs, synaptic transmission, hippocampus, synaptic plasticity, immediate early genes (IEGs)

## Abstract

A tight regulation of the balance between inhibitory and excitatory synaptic transmission is a prerequisite for synaptic plasticity in neuronal networks. In this context, the neurite growth inhibitor membrane protein Nogo-A modulates synaptic plasticity, strength, and neurotransmitter receptor dynamics. However, the molecular mechanisms underlying these actions are unknown. We show that Nogo-A loss-of-function in primary mouse hippocampal cultures by application of a function-blocking antibody leads to higher excitation following a decrease in GABA_A_Rs at inhibitory and an increase in the GluA1, but not GluA2 AMPAR subunit at excitatory synapses. This unbalanced regulation of AMPAR subunits results in the incorporation of Ca^2+^-permeable GluA2-lacking AMPARs and increased intracellular Ca^2+^ levels due to a higher Ca^2+^ influx without affecting its release from the internal stores. Increased neuronal activation upon Nogo-A loss-of-function prompts the phosphorylation of the transcription factor CREB and the expression of c-Fos. These results contribute to the understanding of the molecular mechanisms underlying the regulation of the excitation/inhibition balance and thereby of plasticity in the brain.

## 1. Introduction

A tight balance between excitatory and inhibitory synaptic transmission lies beneath brain function from the developmental stage throughout life. Indeed, the excitation-inhibition (E/I) balance has been shown to shape the response properties of neurons and to regulate their plastic properties [1]. Particularly, the regulation of experience-dependent synaptic plasticity at the onset of the critical period takes place through changes in the GABA-mediated transmission, by these means establishing an adaptive E/I balance [2,3]. Furthermore, alterations of GABAergic transmission are known to underlie several neurological diseases ([3,4] for a review). Thus, understanding the mechanisms controlling the synaptic localization of GABA receptors is extremely relevant. Indeed, the strength of inhibitory synaptic transmission depends on the number of gamma-aminobutyric acid receptors (GABA_A_Rs) present at synapses [5]. The synaptic localization of GABA_A_Rs is, in turn determined both by their clustering via synaptic scaffold proteins as well as by the control of their lateral diffusion through changes in the intracellular Ca^2+^ concentration [6]. Indeed, the number of GABA_A_Rs accumulating at the synapse has been shown to depend on the release of Ca^2+^ from the internal stores via activation of the IP3R1 and PKC [7,8]. Moreover, GABA_A_R diffusion has also been shown to rely upon the Ca^2+^ influx via NMDA receptors followed by the downstream activation of the phosphatase calcineurin and the subsequent dephosphorylation at the Serine-327 of the GABA_A_R γ2 subunit [8,9]. Thus, while the intracellular signaling events regulating the strength of inhibitory synaptic transmission are well described, the molecular mechanisms upstream of the changes in intracellular Ca^2+^ remain so far largely unexplored.

Nogo-A is a membrane-bound growth inhibitor expressed both in oligodendrocytes and neurons [10] of the central nervous system. While its expression decreases strongly during development, it remains elevated in neurons in highly plastic regions within the rodents mature brain, e.g., the hippocampus and cortex [11], indicating its possible role in regulating activity-dependent functional and structural synaptic plasticity (for reviews see [12,13]) both in the hippocampus [14,15,16,17,18,19] and in the cortex [20,21] of rodents. Moreover, published data support the notion, that Nogo-A signaling through its receptors Nogo receptor 1 (NgR1) and Sphingosine-1-Phosphate receptor 2 (S1PR2) reciprocally modulates excitatory and inhibitory synaptic transmission by controlling the delivery at synapses of a-amino-3-hydroxy-5-methyl-4-isoxazole propionic acid receptors (AMPARs; [16,22]) and of GABA_A_Rs ([23,24] for a review) in mouse hippocampal neurons. Specifically, loss-of-function for Nogo-A results in the increased insertion of the GluR1 subunit of AMPARs at synapses as well as a reduced synaptic localization of GABA_A_Rs. The loss of synaptic GABA_A_Rs is due to the increase in their membrane diffusion and is associated with a reduction in the amplitude of GABAergic miniature inhibitory post-synaptic currents in mouse CA3 pyramidal neurons [23]. The regulation by Nogo-A of GABA_A_R diffusion is associated with an increase in the intracellular Ca^2+^ concentration and requires the activation of calcineurin followed by the dephosphorylation of the GABA_A_R γ2 subunit [23]. These observations indicate that Nogo-A regulates the fine-tuning of the E/I balance in the hippocampus. However, the mechanisms mediating this action of Nogo-A remain unknown. Moreover, the source of the increased Ca^2+^ signals observed in pyramidal hippocampal neurons upon Nogo-A neutralization is still unidentified. We show here that blocking Nogo-A in pyramidal hippocampal neurons increases neuronal excitability at the single cell level. Furthermore, we report that release of Ca^2+^ from internal stores is not involved downstream of Nogo-A, despite the large amount of Nogo-A located in the ER. Further on, we report here, that the effect of Nogo-A in regulating inhibitory synaptic strength and neuronal activation depends upon the influx of extracellular Ca^2+^, via a shift in the subunit composition toward a higher presence of Ca^2+^-permeable AMPARs at synapses.

## 2. Materials and Methods

### 2.1. Animal Procedures

All procedures concerning animals were approved by the animal welfare representative of the TU Braunschweig and the LAVES (Oldenburg, Germany, Az. §4 (02.05) TSchB TU BS).

### 2.2. Primary Mouse Hippocampal Culture

Dissociated cultures of mouse hippocampal neurons were prepared from C57Bl/6 mice at embryonic day 18 as previously described [16,19]. The pregnant mouse was killed by cervical dislocation, the mouse embryos were rapidly extracted from the uterus and immediately decapitated. The brain was dissected under sterile conditions and kept in ice-cold Gey’s balanced salt solution (GBSS containing in mM 1.5 CaCl_2_*2H_2_O, 5 KCL, 0.22 KH_2_PO_4_, 1 MgCl*6H_2_O, 0.28 MgSO_4_*7H_2_O, 137 NaCl, 2.7 NaHCO_3_, 0.86 NaH_2_PO_4_) supplemented with 5.5 mM D-Glucose and adjusted to a pH 7.2. The hippocampi were dissociated first by incubation with Trypsin/EDTA at 37 °C for 30 min and subsequently by mechanical dissociation by passing them through a Pasteur pipette. The neurons were plated on poly-L-lysine coated glass coverslips at a density of 3.5 × 10^4^ or 7 × 10^4^ cells per well, depending on the experimental use. The neurons were kept in Neurobasal medium (NB^−^, #21103049, Thermo Fisher, Waltham, MA 02145, USA) supplemented with 2% B27, 11% N_2_, and 0.5 mM Glutamax (NB^+^) at 37 °C, 5% CO_2_, and 99% humidity. The culture medium was partially (20%) exchanged weekly until the cultures were used for experiments at day 21–25.

### 2.3. Organotypic Hippocampal Slice Culture

For whole cell patch clamp recordings, organotypic hippocampal slice cultures were prepared from postnatal day 5 (P5) C57Bl/6 mice of either sex as described previously [25,26]. The mice were swiftly decapitated, the hippocampi were dissected and transferred into ice-cold sterile GBSS supplemented with glucose and adjusted to a pH 7.2. Transversal slices of the hippocampus were cut at a thickness of 400 µm using a McIlwain tissue chopper and let to recover for 30 min in GBSS at 4 °C. The slices were then placed on Millicells CM membrane inserts (Millipore, Merck KGaA, Darmstadt, Germany) and cultivated at 37 °C, 5% CO_2_, and 99% humidity in a medium composed of 50% BME (Eagle, with Hanks salts without glutamine), 25% Hank’s Buffered Salt Solution (HBSS containing 50 mL HBSS (10×), 175 mg NaHCO_3_, 147 mg CaCl_2_*2H_2_O, 1351 mg Glucose, total volume of 500 mL), 1% glucose, 25% donor equine serum (HyClone), and 0.5% L-glutamine. After 72 h, a mixture of antimitotic drugs (cytosine arabinoside, uridine, and fluorodeoxyuridine; stock solution of 1 mM each) was applied at a final concentration of 1 µM each to reduce the number of non-neuronal cells. After 24 h, the medium was fully changed once and then 50% of it every 7 days until the slices were used for experiments at DIV21.

### 2.4. Method Details

#### 2.4.1. Antibody and Peptide Treatment

Nogo-A loss-of-function was achieved via the application of the Nogo-A-NiG specific, monoclonal function-blocking antibody 11C7. The 11C7 antibody is raised against a 18-aa peptide located in the inhibitory domain at the N-terminus of Nogo-A (11C7, mouse IgG1; [27,28,29]; gift from Novartis Pharma AG, Basel, Switzerland to Martin Schwab). A mouse IgG1 against anti-BrdU (FG12, gift from Novartis Pharma AG, Basel, Switzerland to Martin Schwab) was used as control for the Nogo-A loss-of-function experiments. Both antibodies were diluted to a final concentration of 5 µg/mL in the different treatment solutions, depending on the experimental set-up as described below.

#### 2.4.2. Patch Clamp Electrophysiology

Somatic whole-cell recording was performed for pyramidal neurons in the CA3b area of 21 to 25 DIV organotypic mouse hippocampal slice cultures. To this aim, the slice cultures were transferred to an open imaging chamber at a constant temperature of 32 °C continuously perfused (1 mL/min) with oxygenated Artificial Cerobrospinal Fluid (ACSF containing in mM 125 NaCl, 2.5 KCl, 26 NaHCO_3_, 1.25 NaH_2_PO_4_, 2 MgCl_2_*6H_2_O, 2 CaCl_2_*2H_2_O, 25 D-glucose*H_2_O; pH 7.4) and supplemented with 1 µM tetrodotoxin (TTX) to prevent the generation of action potentials. After a 20 min adaptation, the pyramidal neurons were identified visually under the microscope (Axioskop 2 FS Plus, Zeiss, Oberkochen, Germany) using a water-immersion objective (40 × 0.8 NA) and patched. For whole-cell patching, glass pipette electrodes were pulled with a PC-10 vertical micropipette puller (Narishige, Tokyo, Japan) from borosilicate capillaries (1.5 mm).

For action potential threshold and frequency recordings, the capillaries were filled with an internal solution compose of in mM: 126 K-gluconate, 4 KCl, 10 HEPES, 4 MgATP, 0.3 NaGTP, 10 Na_2_phosphocreatine; pH 7.3; 290 mOsm. The resistance of the pipette electrode ranged between 3–7 MΩ. Neurons were patched in current clamp mode and current was injected to hold the cell potential at ∼−70 mV. Two distinct current injection protocols were applied every 5 min for up to 20 min: (1) to assess the action potential threshold, 10 pA steps (100 ms duration, 2 s intervals) were injected until an action potential was generated. (2) Action potential frequency was determined by the injection of 100 pA steps (400 ms duration, 2 s intervals) up to 1 nA.

For recording mEPSCs and mIPSCs simultaneously from individual pyramidal neurons, the pipette electrode was filled with an internal solution composed of: (in mM) 0.5 CaCl_2_, 130 CsCl, 5 EGTA, 30 HEPES, 1 MgCl_2_, and 5 NaCl (pH 7.3). Neurons were voltage-clamped at −50 mV and mPSCs were recorded every 5 min for 120 sec. The recorded amplitudes were analyzed and separated according to their fast (AMPAR) and slow (GABAR) decay time. To define the decay time for mEPSCs and mIPSCs, a preliminary set of experiments was performed by pharmacologically blocking of mEPSCs using CNQX (10 µM) and AP5 (50 µM) and of mIPSCs using bicuculline (10 µM) for the duration of the recording (Appendix A). According to the results of their pharmacological characterization, mPSCs with a decay <25 ms were classified as mEPSCs and those with decay >25 ms as mIPSCs. In loss-of-function experiments, the mPSCs were recorded once before and every 5 min after starting antibody application. Input resistance (R_in_) and series resistance (R_s_) were monitored throughout the recordings and only stable cells (<25% change in R_in_ and R_s_) with R_in_>100 MΩ and R_s_ < 25 MΩ were included in the analysis. Signals were amplified using a Multiclamp 700B amplifier (Molecular Devices, San Jose, CA 95134, USA) and digitized with a Digidata 1550B digitizer (Molecular Devices). The data analysis was performed with Mini Analysis software (Justin Lee). Statistical analysis was performed in Prism (GraphPad, San Diego, CA 92108, USA) using a Repeated measures (RM) Two-Way ANOVA with Fisher’s LSD post hoc test for multiple comparisons.

#### 2.4.3. Live-Cell Labeling and Immunocytochemistry

Live-cell labeling of surface neurotransmitter receptors was performed in 21–25 DIV primary mouse hippocampal cultures plated at a low cell density of 35,000 neurons per coverslip to be able to easily image isolated dendrites. The neurons were incubated with the primary antibody for surface receptor labeling diluted in NB^−^ medium containing 1% BSA at 37 °C, 5% CO_2_, and 99% humidity. The antibody treatment was performed using the Nogo-A or the control antibody at the stated concentrations (see antibody and peptide treatment section). Primary antibodies specific for the extracellular domain of GABA_A_ and AMPA receptors, respectively, were used to assess their surface localization. The anti-GABA_A_ receptor antibody anti-GABA_A_R γ2 (Alomone Labs, Cat# AGA-005, 1:500) was added 20 min before the 10-min antibody treatment. In the case of the AMPA receptors, the anti-AMPAR 1 GluA1 (Alomone Labs, Jerusalem, Israel, Cat# AGP-009, 1:50) and anti-AMPAR 2 GluA2 (Alomone Labs, Cat# AGC-005, 1:50) were co-applied with the Nogo-A or control antibodies for 10 min. After completion of the treatment, the coverslips were rinsed with pre-warmed NB^-^ medium and fixed with 4% paraformaldehyde (PFA) in phosphate buffer (PB containing in mM 50 NaH_2_PO_4_*2H_2_O, 85 Na_2_HPO_4_*2H_2_O) for 10 min at room temperature (RT). After washing them with phosphate buffered saline (PBS containing in mM 1.5 KH_2_PO_4_*2H_2_O, 2.7 KCl, 137 NaCl, 10.4 Na_2_HPO_4_*2H_2_O), the neurons were permeabilized with 0.3% Triton X-100 in PBS for 5 min and successively the unspecific binding sites were blocked with 2 BSA in PBS for 30 min. Subsequently, the neurons were incubated with an anti-synapsin 1/2 antibody (Chicken anti-Synapsin1/2, Synaptic Systems, Göttingen, Germany, Cat# 106006, 1:1000) diluted in PBS containing 2% BSA for 1 h followed by the incubation with the following secondary antibodies (1:500, Jackson Laboratories, Bar Harbor, ME 04609, USA): anti-rabbit Cy2 (#111-225-144), anti-rabbit Cy3 (#111-165-144), anti-chicken Alexa Fluor^®^ 488 (#703-545-155), anti-guinea pig Cy3 (#706-166-148) in PBS for 40 min. Finally, the neurons were mounted with Fluoro-Gel (Electron Microscopy Sciences, Hatfield, PA 19440, USA) onto glass slides for imaging.

The immunocytochemistry for c-Fos and pCREB as markers for neuronal activity was performed in primary hippocampal neurons with a cell density of 70,000 neurons per coverslip. After a 10 min application of either Nogo-A or control antibody in NB^+^ medium, the neurons were fixed with 4% PFA in PB for 10 min at RT. For experiments in which pCREB was labelled, the neurons were fixed immediately after the end of the treatment, while for those in which c-Fos was stained, the cells were still kept 80 min in their original NB^+^ medium before fixation. The neurons were incubated in a permeabilizing and blocking solution containing 0.2% Triton-X 100 and 1.5% goat or donkey serum in PBS for 1 h on the shaker and afterwards with the primary antibodies anti-pCREB (Cell-Signaling, Danvers, MA, 01923, USA, Cat# 9198, 1:500) or anti-c-Fos (Synaptic Systems, Cat# 226003, 1:10,000) diluted together with anti-MAP2 (Abcam, Cambridge, UK, Cat# ab24640, 1:500) in the blocking solution over night at 4 °C. After washing the neurons, the secondary antibodies (1:500, Jackson Laboratories), anti-mouse Cy2 (#115-225-146), and anti-rabbit Cy3 (#111-165-144) were diluted in PBS and added to the neurons for 2 h at RT. Finally, the neurons were incubated for 5 min with Dapi (1:1000) and mounted with Fluoro-Gel (Electron Miscroscopy Sciences) onto glass slides for imaging.

#### 2.4.4. Widefield Fluorescence Imaging and Analysis

2D fluorescence images were acquired using an upright Axio Imager M2 microscope (Zeiss) equipped with an oil-immersion objective (63× NA 1.4) and a CCD camera. Primary dendrites of fluorescently labeled and isolated neurons were chosen, based on the presynaptic marker Synapsin 1/2, which should ensure comparability of neurons and their health. For all experiments in which the surface neurotransmitter receptors were labelled, neurons from a single culture preparation were imaged with a constant sub-saturation exposure time. Both the imaging and analysis were performed blind to the treatment. For the analysis, background fluorescence of 2D images was determined in ImageJ (National Institutes of Health) by placing Regions of Interests (ROIs) in areas of the dendrite of interest in which no specific staining was visible. The mean grey value of all ROIs per dendrite was averaged and defined as background fluorescence in the further analysis. To determine puncta density, area and fluorescence intensity of synaptic proteins, the calculated background was used to set a threshold for each specific dendrite. The analysis was performed using the software SynPAnal [30] with either two-fold subtraction of the background for anti-synapsin 1/2 or GABA_A_ receptors or three-fold subtraction of the background for the AMPA receptors. The colocalization of surface neurotransmitter receptor immune-positive puncta with synapsin 1/2-positive puncta was defined by creating an artificial colocalization channel using ImageJ. The individually determined thresholds per dendrite for each channel were set separately and the images were afterwards color merged. The obtained overlaying puncta were in turn analyzed for cluster density in SynPAnal.

For c-Fos and pCREB analysis, the imaging was performed using a 10× objective (NA 0.3). A total of 7–10 fields of view containing comparable numbers of neurons were randomly chosen for imaging based on the MAP2 fluorescent staining. For each experimental repetition, the light intensity and exposure time was kept constant for all conditions. For the analysis of immune-positive neurons, a threshold was set based on the average fluorescent intensity measured from 10 neurons under control conditions. The positive neurons for pCREB and c-Fos were counted and given as the percentage of all MAP2-labeled neurons within the field of view.

#### 2.4.5. Ca^2+^-Imaging

Primary mouse hippocampal neurons were transfected with the genetically encoded calcium indicators (GCaMP) GCaMP5g [31] or ER-GCaMP6-150 [32] using Lipofectamine 2000 (Invitrogen) at DIV20–25. After 24 h for GCaMP5g and 72 h for ER-GCaMP6-150, the coverslips were transferred to a recording chamber filled with HBSS at room temperature and were let to adjust for 20 min. By using a peristaltic pump with a constant speed of 0.8 mL/min, a continuous flow of HBSS was achieved. Live-cell imaging of randomly chosen GCaMP expressing neurons was performed using a fluorescence microscope (Olympus BX61WI) equipped with a 40× objective (LUMPLFLN W, NA 0.7) and a CCD camera (VisiCam QE, Visitron Systems). Time lapse recordings were acquired using XCellence pro imaging software with a binning of 4 × 4 (336 × 256). To monitor the fluorescence intensity changes and avoid bleaching, a light intensity of 23% and exposure time of 83 sec were set and kept constant throughout the whole experiment.

Using GCaMP5g, time lapse recordings of 500 frames at 5 Hz were acquired. The transfected neurons were imaged twice before the treatment with either the Nogo-A or the control antibody to record the baseline condition. Ten minutes after starting antibody application, the neurons were recorded once again to monitor the effect of the Nogo-A loss-of-function. Immediately afterwards, the specific blockers for NMDARs (AP5, 50 µM) and AMPARs (NBQX, 5 µM; Naspm, 5 µM) were washed-in and again after another 10 min, the neuron were monitored.

For ER-GCaMP6-150 experiments assessing the changes in Ca^2+^ within the ER, neurons were imaged in a two-part experiment. Initially, 6000 images with a cycle time of 0.2 sec (5 Hz) were recorded. The antibody treatment was started after an initial imaging time of 5 min and the time lapse recordings were continued for another 15 min. For the second part of the experiment, 2700 images with a cycle time of 1 sec (1 Hz) were recorded. During the entire time course of the experiment (45 min), the Nogo-A or control antibody was present in the circling HBSS. Additionally, after 5 min of recording the SERCA-ATPase blocker Cyclopiazoic acid (CPA, 1 µM) was added for a total time of 5 min and subsequently washed out.

For analysis, the regions of interest (ROIs) were determined using ImageJ software and analyzed with a MatLab-based self-established protocol. For background correction, an additional ROI was drawn on an area with no fluorescence, representing resting activity. The ROIs were placed on the cell body, and for GCaMP5g experiments, additionally on dendritic spines. To evaluate changes in Ca^2+^, the amplitude and frequency of the fluorescence maxima were detected. In the case of the recorded dendritic spines, the individual ROIs were averaged per cell. To calculate the change in fluorescence intensity, the following equation was used: ΔF/F_0_ = [(F − B) − (F_0_ − B_0_)]/(F_0_ − B_0_), where F_0_ and B_0_ represent the mean grey value of the selected ROIs at resting conditions. In transfected neurons with ER-GCaMP6-150, the calculation makes it possible to determine the resting condition and to distinguish negative and positive amplitudes, reflecting the status of Ca^2+^ binding. To compensate for the bleaching, the recorded Ca^2+^ traces were filtered and detected by a MatLab in-house tool. Data were normalized to the first time point before treatment because 2 records were acquired before antibody treatment. With respect to the ER-GCamP5g experiments with CPA, wash-in raw data were plotted. For statistical analysis, a repeated-measures ANOVA with a Fisher’s LSD post hoc test was performed.

## 3. Results

### 3.1. Nogo-A Loss-of-Function Shifts the Excitation/Inhibition Balance toward a Higher Excitation at a Single Cell Level

Nogo-A has been shown to confine the recruitment of excitatory postsynaptic α-amino-3-hydroxy-5-methyl-4-isoxazolepropionic acid receptors (AMPARs) at the membrane surface [16,22], thereby regulating the strength of glutamatergic synaptic transmission. In addition, Nogo-A signaling promotes inhibitory signaling by regulating the synaptic localization of γ-aminobutyric acid type A receptors (GABA_A_Rs; [23]) to strengthen inhibitory synaptic transmission. These data indicate that Nogo-A signaling reciprocally regulates excitatory and inhibitory synaptic transmission, thus influencing the E/I balance. However, the cellular mechanism and whether the shift in E/I balance upon Nogo-A loss-of-function occurs at a network or at a single-cell level remains unexplored. This question was addressed here by analyzing miniature postsynaptic currents (mPSCs) in hippocampal organotypic mouse slice cultures using whole-cell patch clamp to simultaneously record the mEPSCs and mIPSCs of an individual CA3 pyramidal neuron. We took advantage of the different decay kinetics of glutamatergic mEPSCs and GABAergic mIPSCs [33] and classified mPSCs with decay kinetics below 25 ms as mEPSCs and above 25 ms as mIPSCs (Figure 1A). This classification was confirmed via the application of specific pharmacological blockers (Appendix A) during the recording of mPSCs to distinguish mEPSCs (CNQX and AP5) from mIPSCs (bicuculline). Comparing the ratio of the amplitude of mEPSCs and mIPSCs for each neuron before and after application of either the control or the Nogo-A function-blocking antibody, a significant shift was observed towards a higher net excitation (Figure 1B; Appendix A). While the average mIPSC/mEPSC ratio for each neuron remained stable over time relative to the first time point upon application of control antibody, acute neutralization of Nogo-A led to a significant reduction of the mIPSC/mEPSC ratio both after 5 and after 10 min of treatment (Figure 1A,B; RM 2-way ANOVA _treatment × time_, F_(2,52)_ = 11.44, *p* < 0.001, Fisher’s LSD post-hoc test). Comparing the average ratio between mEPSCs and mIPSCs after 10 min of treatment with the control or the Nogo-A function-blocking antibody for each individual neuron showed a net shift in E/I balance toward more excitation and less inhibition (Figure 1C; Appendix A). When normalized, the values obtained upon Nogo-A neutralization reveal an increase of mEPSC amplitude of ∼7% (Nogo-A Ab: 6.07% ± 4.0; Ctrl Ab: −0.85% ± 3.9) and a decrease in mIPSC amplitude of ∼8% (Nogo-A Ab: −8.53% ± 1.3; Ctrl Ab: −0.57% ± 1.1) compared to the control condition.

While the data so far indicated a shift toward a higher excitation upon Nogo-A neutralization, they leave open the question of whether this effect observed at the single cell level derives from a change only in excitation, only in inhibition or in both. Thus, to clarify the mechanism underlying the E/I balance shift at the level of individual neurons, the percent change in the mIPSCs and the mEPSCs amplitude was plotted for each neuron before and after the addition of the control or the Nogo-A blocking antibody (Figure 1D,E; Appendix A). After 5 and 10 min of treatment with control antibody, the amplitude of both mEPSCs and mIPSCs showed only a minor, non-significant deviation from the baseline relative to the pre-treatment time point (Figure 1D). In contrast, only 5 min of Nogo-A neutralization already affected the amplitude of both mEPSCs and mIPSCs for all recorded cells. Particularly, a significant reduction in the mIPSC amplitude and a significant increase in the amplitude of mEPSC could be observed. A comparable effect was seen when the loss-of-function lasted for 10 min (Figure 1E; RM 2-way ANOVA _mPSCs × time_, F_(2,52)_ = 11.27, *p* < 0.001, Fisher’s LSD post-hoc test). Next, the change in amplitude relative to the time point before treatment was plotted in percent for the individual cells over time in a heat map (Figure 1F,G). Under control conditions after 5 min of control antibody application, ∼21% of the pyramidal neurons reacted above average with a decrease in the mIPSC and ∼21% with an increase in mEPSC amplitude (Figure 1F). After 10 min of control treatment, ∼28% of the pyramidal neurons showed a decrease in mIPSC amplitude, while only ∼14% showed an increase in the amplitude of mEPSCs (Figure 1G). Interestingly, upon Nogo-A antibody treatment, ∼79% of the neurons showed a reduction in mIPSC amplitude already after 5 min (Figure 1F). At the same time, an increase in mEPSC amplitude can be observed in ∼50% of the neurons when Nogo-A is blocked (Figure 1G). The size of these effects remained stable also after 10 min of Nogo-A neutralization (Figure 1F,G). It is striking that while for some neurons an opposite shift could be observed in mIPSCs and mEPSCs (cells 1,3, 7, 9 11, 12, and 14), most of the neurons respond to the Nogo-A neutralization by regulating only the amplitude of the mIPSCs (cells 2, 4–6, 10, and 13) and only one of the mEPSCs (cell 8). These observations indicate that the shift in the E/I balance observed upon Nogo-A neutralization depends upon a complex regulation of excitatory and inhibitory synaptic strength at the single cell level. Moreover, it can be observed that at this time window, the reduction of mIPSC amplitude upon Nogo-A loss-of-function seems to precede the increase in mEPSC amplitude. Indeed, although the change in percent is smaller than the one of the mEPSCs, more neurons initially respond with a decrease in mIPSC amplitude upon Nogo-A loss-of-function (Figure 1F,G).

The strength of excitatory and inhibitory synaptic transmission correlates with the number of specific receptors at postsynaptic sites [34,35,36]. Indeed, we recently showed, in two separate series of experiments that Nogo-A signaling regulates the recruitment of AMPARs and the confinement of GABA_A_Rs, respectively, at synaptic sites [23]. Here, a similar shift in the synaptic localization of GABA_A_Rs and AMPARs can also be observed within individual neurons. The synaptic localization of postsynaptic neurotransmitter receptors was analyzed by live-cell co-labeling of the AMPAR GluA1 extracellular domain and GABA_A_Rs in primary hippocampal cultures (Figure 1H). A shift in surface receptor balance towards more excitatory AMPARs could be seen when Nogo-A was blocked, resulting in a global alteration of the average GABA_A_R/GluA1 ratio for individual neurons (Figure 1H,I; Unpaired Student’s *t*-Test, t = 4.729, df = 79, *p* < 0.001; Appendix A). After 10 min Nogo-A loss-of-function, the normalized data show an increase in surface AMPAR GluA1 by ∼21% (Nogo-A Ab: 21.0% ± 5.1; Ctrl Ab: 0% ± 5.0%) and a decrease in postsynaptic clustered GABA_A_R by ∼28% (Nogo-A Ab: −28.05% ± 4.2; Ctrl Ab: 0% ± 4.7) along individual dendrites of single neurons (Figure 1J; Appendix A).

Taken together, the results so far indicate a role of Nogo-A in regulating the E/I balance within the hippocampal CA3 region by a complex modulation of the strength of the excitatory and inhibitory synaptic transmission at the single cell level.

### 3.2. Nogo-A Regulates the Synaptic Insertion of Calcium Permeable-AMPARs

The synaptic localization and trafficking of AMPARs are key determinants of the strength of excitatory transmission [37]. In this context, the subunit composition of AMPARs and especially their content of GluA1 and GluA2 AMPAR subunits play a major role [38,39,40]. Interestingly, a transient incorporation of GluA2-lacking AMPARs was shown during the first 10 min after LTP induction, suggesting that the postsynaptic integration of calcium-permeable AMPARs (CP-AMPARs) provides additional signals to support the initial activity-dependent processes [41,42]. As we have previously shown, Nogo-A restricts long-term potentiation (LTP, [14]) and affects Ca^2+^ dynamics [23]. Indeed, its acute loss-of-function results in a higher LTP in acute hippocampal slices and in a significant increase in the amplitude of Ca^2+^ transients in primary hippocampal neurons [23]. This observation suggests a possible role of Nogo-A in regulating not only GluA1 insertion, but also GluA2 surface localization. To address this question, each of the two AMPAR subunits was labeled in living primary hippocampal neurons followed by post-hoc labelling for the presynaptic marker synapsin 1/2 (Figure 2A,B). First, the synaptic localization of GluA1 subunit was analyzed upon Nogo-A loss-of-function. After 10 min of Nogo-A neutralization, the density of GluA1 immuno-positive clusters increased significantly by ∼14% (Figure 2A,C; Unpaired Student’s *t*-Test, t = 2.331, df = 77, *p* = 0.022; Appendix A). In addition, there is an enlargement in the area by ∼10% of GluA1 immuno-positive clusters compared to the control condition (Figure 2A,D; Unpaired Student’s *t*-Test, t = 2.298, df = 77, *p* = 0.024; Appendix A). Further, the density of GluA1-positive clusters colocalized with synapsin 1/2-positive puncta, was significantly enhanced by ∼11% (Figure 2A,E; Unpaired Student’s *t*-Test, t = 2.101, df = 77, *p* = 0.039; Appendix A). These results indicate an increase in GluA1 at active synaptic sites after 10 min Nogo-A antibody treatment. In contrast, the acute neutralization of Nogo-A did not alter the surface localization of the GluA2 subunit. No detectable difference could be observed in the density or area of the immuno-positive GluA2 clusters between the treatment with Nogo-A function blocking and control antibodies (Figure 2B,F,G; Appendix A). Likewise, the colocalization of GluA2-positive clusters with synapsin 1/2-positive puncta showed no difference (Figure 2B,H; Appendix A).

These results strengthen the previous observations that Nogo-A regulates the insertion of GluA1 at the postsynaptic membrane and additionally show that this effect already occurs after 10 min of Nogo-A loss-of-function. Furthermore, they indicate that Nogo specifically regulates the synaptic insertion of GluA1, but not of the GluA2 subunit. This suggests a specific increase in the proportion of CP-AMPARs at synaptic surfaces after 10 min of Nogo-A neutralization.

### 3.3. Nogo-A Neutralization Results in a Calcium Permeable-AMPAR Dependent Increase in Ca^2+^ Dynamics

To investigate whether the relative increase in the surface localization of CP-AMPARs observed upon Nogo-A neutralization is required for the modulation of Ca^2+^ dynamics, we recorded Ca^2+^ transients in primary hippocampal neurons by using a genetically encoded Ca^2+^ indicator (GCaMP5g). The neurons were treated with either Nogo-A function-blocking or control antibodies for 10 min and additionally with different pharmacological blockers for AMPARs (NBQX, 5 µM), CP-AMPARs (Naspm, 5 µM), and *N*-Methyl-D-Aspartate (NMDA) receptors (APV, 50 µM) for 10 more min (Figure 3A). A 10 min Nogo-A neutralization resulted in a two-fold increase in the amplitude of the Ca^2+^ transients. However, when in addition to the antibody treatment, the selective AMPAR blocker NBQX was washed-in, the amplitude of Ca^2+^ transients was rapidly and significantly decreased for both control and Nogo-A antibody-treated neurons (Figure 3B; RM 2-way ANOVA _time × treatment_, F_(2,40)_ = 5.217, *p* < 0.01, Fisher’s LSD post-hoc test; Appendix A). Thus, no difference can be observed between the two antibody conditions in the fluorescence intensity level relative to baseline upon blocking the AMPARs (Nogo-A Ab: 30% ± 6,53; Ctrl Ab: 35% ± 7.88) (Figure 3B; RM 2-way ANOVA _time_, F_(1.35,27.04)_ = 13.98, *p* < 0.001, Fisher’s LSD post-hoc test; Appendix A). This effect was also observed for the frequency of Ca^2+^ dynamics (Figure 3C; RM 2-way ANOVA _time_, F_(1.22,24.22)_ = 13.98, *p* < 0.001, Fisher’s LSD post-hoc test; Appendix A). To better elucidate the involvement of AMPARs in the increase in Ca^2+^ dynamics upon Nogo-A loss-of-function, next only CP-AMPARs were blocked by the selective antagonist Naspm. While the Ca^2+^ transients of neurons treated with Nogo-A blocking antibodies showed a significant rise in amplitude (Figure 3D; RM 2-way ANOVA _treatment × time_, F_(2,58)_ = 3.456, *p* = 0.038, Fisher’s LSD post-hoc test; Appendix A), when the neurons were additionally treated with Naspm, this increase in Ca^2+^ amplitude was completely abolished and the fluorescence intensity level was reduced by ∼5% relative to the baseline (Nogo-A: 95% ± 8.58). Under control conditions, the presence of Naspm led to a decrease of the Ca^2+^ transient amplitude by ∼20% (Ctrl Ab: 79% ± 18.73) (Figure 3D; RM 2-way ANOVA _time_, F_(1.08,31.21)_ = 5.429, *p* = 0.024, Fisher’s LSD post-hoc test; Appendix A). No statistically significant difference could be observed between the two antibody conditions. The frequency of Ca^2+^ transients was not affected, neither the antibody treatment nor by the application of Naspm (Figure 3E; Appendix A).

Nogo-A signaling has been shown to affect, in addition to GluA1 and GluA2, also the expression of NMDAR subunits in hippocampal neurons [43]. Therefore, next, the effect of a possible acute regulation of NMDARs was examined regarding the dynamics of Ca^2+^ influx upon Nogo-A neutralization. A total of 10 min of Nogo-A neutralization resulted in a significant, 2-fold increase in Ca^2+^ amplitude compared to the control condition (Figure 3F; RM 2-way ANOVA _treatment × time_, F_(2,48)_ = 5.687, *p* = 0.006, Fisher’s LSD post-hoc test). In contrast, when the competitive NMDAR antagonist APV was washed-in, the increase in amplitude of Ca^2+^ transients seen upon Nogo-A blocking antibody treatment was completely abolished (Figure 3F; Appendix A). The fluorescence intensity was equally significantly lower than baseline for both antibody treatments (Figure 3F; Nogo-A Ab: 25% ± 6.80; Ctrl Ab: 30% ± 15.95). Furthermore, the blocking of NMDARs caused an equally significant decrease in the frequency of Ca^2+^ transients under both antibody conditions (Figure 3G; RM 2-way ANOVA _time_, F_(1.63,39.07)_ = 8.922, *p* = 0.001, Fisher’s LSD post-hoc test; Appendix A).

These results suggest an AMPAR and NMDAR-dependent increase in Ca^2+^ influx upon Nogo-A loss-of-function in primary hippocampal neurons, presumably due to the absence of GluA2-containing receptors at the synaptic membrane. Together with live-cell labeling of GluA1 and GluA2, this strengthens the hypothesis that Nogo-A regulates CP-AMPAR insertion at synapses.

### 3.4. Ca^2+^ Influx Versus Intracellular Ca^2+^ Release upon Nogo-A Loss-of-Function

Ca^2+^ dynamics are crucial for synaptic transmission and affect both inhibitory and excitatory signaling. We recently showed that Nogo-A modulates the motility and synaptic localization of inhibitory postsynaptic GABA_A_R by regulating Ca^2+^ dynamics within primary hippocampal neurons [23]. Moreover, we could provide evidence suggesting a role of the extracellular Ca^2+^ influx in the regulation of post-synaptic neurotransmitter receptors. To take this evidence further and exclude the endoplasmic reticulum (ER) as a possible source of the increased intracellular Ca^2+^ levels, we investigate whether Nogo-A signaling may affect the release of Ca^2+^ from the ER. In this respect, primary hippocampal neurons were transfected with the genetically encoded ER-targeted Ca^2+^ sensor ER-GCaMP6-150 and treated with either a Nogo-A function-blocking or a control antibody. Due to the low affinity properties of ER-GCaMP6-150, Ca^2+^ dynamics were detected in time-lapse imaging experiments (Figure 4A) via changes in fluorescence intensity [32]. Negative and positive changes in fluorescence intensity indicate ER-dependent Ca^2+^ efflux and influx, respectively (Figure 4B). To measure the involvement of internal stores downstream of the Nogo-A signaling, the frequency and amplitude of fluorescence intensity changes were analyzed upon its acute neutralization. In transfected neurons treated with a Nogo-A function-blocking antibody, a higher amplitude of the positive change in fluorescence intensity could be observed over time, becoming significantly higher than under the control conditions after 15 min, by ∼7% (Nogo-A Ab: 11.35% ± 2.77; Ctrl Ab: 4.00% ± 1.73) (Figure 4C; RM 2-way ANOVA _time × treatment_, F_(3,78)_ = 3.131, *p* = 0.030, Fisher’s LSD post-hoc test; Appendix A). Likewise, the frequency of the peaks of positive change of fluorescence intensity increased upon Nogo-A antibody wash-in and showed a consist difference of ∼20% compared to the control antibody, albeit not statistically significant (Figure 4E; RM 2-way ANOVA _time × treatment_, F_(3,78)_ = 1.326, *p* = 0.272; Appendix A). No difference between the two antibody treatments could be observed in either amplitude (Figure 4D) or frequency (Figure 4F; Appendix A) of the negative peaks of fluorescence intensity changes. These results indicate a change in Ca^2+^ transfer through the ER upon Nogo-A loss-of-function, characterized by increased Ca^2+^ entry into the intracellular Ca^2+^ storage sites. To further clarify the impact of the ER in the Ca^2+^ dynamics downstream of Nogo-A, in some experiments the transfer of Ca^2+^ from the cytosol into the lumen of the ER was blocked by inhibiting the SERCA ATPase with a 5 min application of Cyclopiazonic acid (CPA; 10 µM) resulting in a reversible depletion of ER Ca^2+^ stores [44]. Both the neurons treated with control or Nogo-A blocking antibody showed a similar initial fluorescence intensity, which after wash-in of CPA progressively decreased to ∼20% of the baseline within 5–10 min. Upon CPA wash-out, a gradual increase in fluorescence intensity could be observed indicating the re-uptake of Ca^2+^ ions into the ER. Interestingly, the fluorescence intensity curve under control conditions showed a significantly smaller increase than the one of neurons treated with Nogo-A blocking antibodies. While upon Nogo-A neutralization, the fluorescence intensity reached its maximum recovery after 15 min and remained at a plateau of −20% of the baseline, under control conditions the increase was slower reaching −30% of the baseline after 35 min of CPA washout (Figure 4G; RM 2-way ANOVA _treatment_, F_(1,15)_ = 6.382, *p* < 0.001).

In summary, when comparing the two antibody conditions, no difference could be seen in the Ca^2+^ efflux from the ER. This conclusion is supported by the lack of difference in the course of the fluorescence intensity curve during the decreasing phase after depletion of ER Ca^2+^ stores by application of CPA. This suggests an ER-independent increase in Ca^2+^ dynamics upon Nogo-A loss-of-function. This hypothesis is strengthened by the significant difference in the replenishment of internal stores after CPA wash-out and the higher speed and amplitude of the positive changes in the fluorescence intensity curve. Thus, upon treatment with the Nogo-A function-blocking antibody, an increase can be observed in the concentration of cytosolic Ca^2+^, which is recycled through a higher reuptake by the ER.

### 3.5. Nogo-A Loss-of-Function Increases Neuronal Excitability and Neuronal Activation

Next, we assessed whether the regulation of synaptic properties upon Nogo-A loss-of-function might influence neuronal excitability. Based on the previous observation that Nogo-A affects synaptic transmission, we additionally examined the effect of Nogo-A loss-of-function on the threshold and the frequency of evoked action potentials (AP). To this aim, the percentual change of the evoked AP threshold was recorded for CA3 pyramidal neurons in hippocampal organotypic slice cultures in whole patch-clamp electrophysiological experiments. Already after 10 min of blocking the Nogo-A protein with a function-blocking antibody, the threshold for generating APs was reduced. After 20 min of Nogo-A antibody treatment, there was a significant difference by ∼6% (Nogo-A Ab: −3.4% ± 0.76; Ctrl Ab: +2.4% ± 0.63) between the Nogo-A blocking antibody and the control condition (Figure 1A; RM 2-way ANOVA _treatment × time_, F_(2,38)_ = 3.307, *p* = 0.047, Fisher’s LSD post-hoc test; Appendix A). Moreover, the frequency of APs evoked upon current injection was significantly higher during Nogo-A function-blocking antibody treatment. Indeed, after 10 min of Nogo-A loss-of-function, a current injection of 0.4 nA resulted in a significant increase of the frequency of evoked APs by ∼50% (Nogo-A Ab: +43.1% ± 4.18; Ctrl Ab: −13.9% ± 1.49) when compared to the control antibody (Figure 1B; RM 2-way ANOVA _treatment × current injection_, F_(10,180)_ = 2.587, *p* = 0.006, Fisher’s LSD post-hoc test; Appendix A). These results indicate that already after 10 min of Nogo-A loss-of-function, the excitability of CA3 pyramidal neurons is significantly increased.

The somatic membrane potential reflects the overall input activity of a neuron [45] and can be monitored by recording the changes in intracellular Ca^2+^ concentration. To visualize the increase of neuronal activity, primary hippocampal neurons were transfected with the Ca^2+^ indicator GCaMP5g (Figure 5C) and alteration in intracellular Ca^2+^ concentration were measured at the cell body by analyzing the changes in emitted fluorescence over time before and after treatment with Nogo-A or control antibody. The results show a significant increase in the amplitude of Ca^2+^ signals measured at the cell body of primary hippocampal neurons. The percentage change of the amplitude of Ca^2+^ transients is increased by ∼50% (light red) Nogo-A Ab: 154.2% ± 20.6; (grey) Ctrl Ab: 97.9% ± 14.9) after 10 min Nogo-A loss-of function when compared to the controls (Figure 5D, RM 2-way ANOVA _treatment_, F_(1,47)_ = 7.366, *p* = 0.009, Fisher’s LSD post-hoc test; Appendix A). The significant increase was still visible 20 min after the treatment, albeit not significant. Interestingly, the increase in the amplitude of Ca^2+^ transients observed upon Nogo-A neutralization is completely abolished by the washing in of the CP-AMPAR blocker Naspm during the last 10 min of the experiment (Figure 5D, (red) Nogo-A Ab; (black) Ctrl Ab). This result supports the hypothesis that the increased localization of CP-AMPARs plays a central role in increasing the intracellular Ca^2+^ levels and neuronal activation upon Nogo-A loss-of-function.

In neurons, Ca^2+^ influx trough Ca^2+^-permeable ion channels in the plasma membrane is essential for plasticity and regulates gene expression through the activation of the transcription factor cAMP-responsive element binding protein CREB [46]. Furthermore, CREB is involved in the formation of long-term memories and is therefore an interesting candidate to be modulated by Nogo-A signaling. Indeed, Nogo-A is known to influence activity-dependent synaptic plasticity, as blocking Nogo-A has been shown to result in higher long-term potentiation (LTP; [14]), as well as learning and memory processes [19,21]. To examine the role of Nogo-A signaling on CREB phosphorylation, primary hippocampal neurons were treated with either a Nogo-A function blocking or a control antibody for 10 min. The neurons were subsequently immunostained for pCREB and for the neuronal marker MAP2 (Figure 5E) to assess the relative number of pCREB positive neurons. The number of pCREB-positive neurons was significantly increased by ∼15% after 10 min Nogo-A antibody treatment (Nogo-A Ab: +14.8% ± 6.16; Ctrl Ab: 0% ± 3.08), compared to the control condition (Figure 5F, Student’s *t*-Test, t = 2.143, df = 22, *p* = 0.004; Appendix A). Ca^2+^-dependent phosphorylation of the transcription factor CREB leads to the activation of a number of downstream immediate–early genes including *c-fos* [47]. C-Fos expression is upregulated in response to neuronal activity and in association with learning and memory processes [48]. Thus, the expression levels of c-Fos were analyzed within MAP2-positive neurons (Figure 5G). Upon acute Nogo-A loss-of-function, a significant increase could be observed in the number of c-Fos positive neurons by ∼33% when compared to the control condition (Nogo-A Ab: +32.5% ± 8.15) (Figure 5H; Student’s *t*-Test, *t* = 2.998, df = 14, *p* = 0.001; Appendix A).

Taken together, these results show that Nogo-A loss-of-function increases neuronal excitability and activation as well as the activity-induced expression of different plasticity-related molecules such as pCREB and cFos.

## 4. Discussion

In this study, we addressed the mechanism by which Nogo-A signaling is involved in stabilizing the excitation/inhibition (E/I) balance in mouse hippocampal neurons. We show here that it does so by restricting AMPAR and GABA_A_R dynamics and suppressing neurons excitability in a Ca^2+^-dependent manner. Indeed, loss-of-function for Nogo-A results in a shift in the E/I balance toward more excitation due to the insertion of a higher proportion of Ca^2+^-permeable AMPARs, an increased influx of extracellular Ca^2+^, and the reduction of GABA_A_Rs at synaptic sites. These changes result in a Ca^2+^-dependent increase in the activation of hippocampal neurons followed by the enhanced downstream expression of activity-induced immediate early genes (IEGs).

We recently showed that Nogo-A signaling via its S1PR2 receptor suppresses excitatory and promotes inhibitory synaptic transmission [23] and that it regulates AMPAR insertion at synapses [16] and GABA_A_Rs diffusion along the membrane [23] in an opposite manner. In the current study, we assessed this function of Nogo-A at a single cell level. The results presented here show that Nogo-A loss-of-function increases the strength of excitatory and suppresses the inhibitory synaptic transmission within individual CA3 pyramidal neurons, thereby shifting their E/I balance toward higher excitation. These effects are extremely fast since both excitation and inhibition are altered already by only 5 min of Nogo-A loss-of-function. This activity of Nogo-A is indeed interesting since it possibly underlies its functional role in determining the degree of activity-dependent synaptic plasticity and provides a possible molecular correlate of the regulatory mechanisms existing in neurons to reciprocally regulate the function of excitatory and inhibitory synapses. Fast excitatory/inhibitory transmission is predominantly mediated by ionotropic AMPA/GABA_A_ receptors. This and previously published studies indicate the ability of Nogo-A to regulate neurotransmitter receptor dynamics in mouse hippocampal neurons for review see [24]. Specifically, here we show that Nogo-A loss-of-function promotes the surface expression of GluA1 without affecting GluA2 subunit. An unbalanced regulation in AMPAR subunits is supposed to potentially lead to the generation of GluA2-lacking GluA1 homomers, therefore Ca^2+^-permeable AMPARs (CP-AMPARs), and is also observed following the induction of a homeostatic response [49,50,51]. In support of this idea, our results show an increase in the amplitude of Ca^2+^ transients upon Nogo-A loss-of-function, which is prevented by the application of Naspm, a specific antagonist of CP-AMPARs [49,52]. Importantly, CP-AMPARs have been implicated in hippocampal LTP, where a fast and transient incorporation of GluA2-lacking AMPARs is necessary for the induction and stabilization of LTP [42,53]. Interestingly, blocking Nogo-A signaling via its receptors NgR1 and S1PR2 results in a significant increase in the strength of LTP in different brain regions [14,17,21].

Nogo-A neutralization in hippocampal primary neurons results in the fast and transient increase in the amplitude of Ca^2+^ transients ([23]; this study), which are a requirement for the regulation of inhibitory synaptic strength and the synaptic localization of GABA_A_Rs [23]. Indeed, we recently showed that buffering extracellular Ca^2+^ via the application of the Ca^2+^ chelator EGTA reduces lateral diffusion of GABA_A_Rs after Nogo-A loss-of-function [23]. While this observation indicates that an influx of extracellular Ca^2+^ may underlie the alterations observed in GABA_A_R dynamics, it did, so far, not exclude that release from the internal stores may contribute to the increase in cytoplasmic Ca^2+^ concentration. This question is of relevance since it has been shown that cytoplasmatic Ca^2+^ can have different effects on synaptic regulation depending on where it comes from [6]. While Ca^2+^ release from the endoplasmic reticulum (ER) suppressed GABA_A_R lateral diffusion [8], Ca^2+^ influx through the plasma membrane leads to an increase in their lateral diffusion [8,54]. In this context, it is important to note that Nogo-A is expressed both at the plasma membrane and within the ER [55]. Moreover, the regulation of Ca^2+^ signals as well as of inhibitory synaptic strength upon Nogo-A loss-of-function occurs solely via the S1PR2 [23], which has been shown to affect both the influx of extracellular and its release from the internal stores [56]. Our current study sheds light on this question, since we provide strong evidence that the increase in the amplitude of Ca^2+^ transients after blocking Nogo-A depends exclusively on its influx across the plasma membrane. This claim is based on the Ca^2+^ imaging experiments performed using a genetically encoded ER-targeted Ca^2+^ indicator [32]. This method confirms the increase in intracellular Ca^2+^ concentration. The replenishment of the internal stores occurring after application of the washout of the sarco/endoplasmic reticulum calcium ATPase (SERCA) blocker CPA is significantly faster and stronger in cells treated with Nogo-A blocking antibodies, supporting the increased level of cytoplasmic Ca^2+^ under these conditions. In this context, it is known that the cytosolic Ca^2+^ concentration is kept stable by the SERCA actively pumping Ca^2+^ into the ER [57,58,59]. If SERCA is blocked by CPA application, the ER becomes depleted due to the passive Ca^2+^ efflux from the ER. However, it is important to mention that we cannot completely exclude an extracellular Ca^2+^ influx into the ER by activating store-operated calcium entry (SOCE; [58]). Moreover, the time traces representing ER depleting after perfusion with the SERCA blocker CPA-induced drop in fluorescence intensity as well as its negative peak upon application of Nogo-A function blocking antibodies perfectly overlap with the one recorded in neurons under control conditions, indicating that the depletion of the ER Ca^2+^ store is not affected by the Nogo-A loss-of-function.

Ca^2+^ dynamics are essential for synaptic transmission [60] and adaptive processes in neuronal excitation and inhibition. While upon blocking Nogo-A, most of the CA3 pyramidal neurons show changes both in excitation and inhibition, the effect appears to be faster and stronger in regulating inhibitory synaptic transmission. While at 5 min of treatment, 80% of the cells respond with a decrease in inhibition, only about 50% show a change in excitation. On the other hand, it must be noted that while the changes in inhibition stay stable over the recording time, those in excitation become weaker, suggesting the possibility that the changes at excitatory synapses may precede those at the inhibitory ones, thus occurring at a faster time. Indeed, the diffusion dynamics of GABA_A_Rs at inhibitory synapses and their strength have been shown to be enhanced by an increase in neuronal activity [8], suggesting that changes in excitation drive the alterations at inhibitory synapses. In turn, the activity of inhibitory interneurons has been suggested to play a crucial role during the development of the central nervous system in establishing overall network excitability by normalizing the amount of input arriving within a region [61] as well as by dictating the output of principal neurons [62]. Furthermore, feedback and feedforward inhibition have been suggested to control the total number of active excitatory neurons in the hippocampus, thereby controlling the excitability patterns [63] and regulating the plasticity potential of the neuronal network. Indeed, the activity of Nogo-A in modulating the effect of excitatory transmission on inhibition has important functional implications in the context of activity-dependent synaptic plasticity, since enhancement of activity-dependent neuronal disinhibition [54,64] could in fact be part of the mechanism underlying the increased LTP observed upon Nogo-A loss-of-function [14,17,21]. This is supported by the observation that blocking GABA_A_Rs in Nogo-A function blocking antibody-treated acute slices does not further increase the potentiation at the Shaffer collateral, indicating a possible involvement of GABAergic neurons [14]. In this respect, it is noteworthy that in highly plastic brain regions, Nogo-A as well as its receptor NgR1 not only are expressed in excitatory principal neurons, but also in Parvalbumin-positive (PV+) interneurons [19,65]. Inhibitory interneurons help to maintain the E/I balance following neuronal activation [66,67] as well as regulating activity-dependent synaptic plasticity and, by these means, memory processes [68]. Nogo-A signaling via NgR1 has been shown to act within PV+ interneurons to restrict plasticity in the visual cortex upon monocular deprivation [69] and during erasure of fear memories [70]. Interestingly, deletion of NgR1 specifically from PV+ interneurons has been shown to decrease the excitatory drive on Layer 4 cortical PV+ interneurons, thereby promoting ocular dominance plasticity [71].

The long-lasting enhancement of synaptic strength during plasticity processes requires Ca^2+^ influx and is thought to trigger gene expression, leading to consolidation of synaptic and neuronal changes [72]. The expression of IEGs is widely used as a marker for increased neuronal activation specifically in neuronal populations that undergo plastic changes linked to learning and memory processes [48]. Moreover, the expression of cFos is dependent on the Ca^2+^-dependent activation of the cAMP response element-binding protein (CREB) upon its phosphorylation [73]. In this context, we show here that already after 10 min of Nogo-A, loss-of-function neuronal activation increases, showed by the rise in the amplitude of Ca^2+^ signals and further supported by the increase in CREB phosphorylation and in the expression level of c-Fos in primary hippocampal neurons. Interestingly, a gain-of-function approach via the application of the Nogo-A-Δ20 peptide has been shown to reduce CREB phosphorylation, thereby inducing growth cone collapse in axons of dorsal root ganglion neurons [74]. Additionally, elevated pCREB levels can overcome inhibitors in the myelin and promote spinal cord regeneration [75] as well as neurite inhibition by Nogo-A-Δ20 [74]. The activity and expression of the IEGs CREB [76] and c-Fos [77] are associated with processes resulting in long-term plastic changes at synapses and are the first steps for the formation of long-term memory. In this regard, Nogo-A signaling has been shown to restrict activity-dependent functional and structural synaptic plasticity see for review [12,13] as well as learning and memory processes [19,21].

## 5. Conclusions

Taken together, the results presented here identify Nogo-A as a main player in the molecular mechanisms regulating the E/I balance and synaptic plasticity and define the downstream signaling involved in this activity. Particularly, we propose that Nogo-A signaling stabilizes the E/I balance by controlling the neurotransmitter receptor dynamics in a Ca^2+^-dependent manner. The ability of Nogo-A to modulate neuronal excitability as well as pCREB and c-Fos levels makes it extremely interesting in the context of research addressing the molecular mechanisms regulating learning and memory processes, and particularly the formation of memory engrams in healthy and pathological conditions, like the implementation of unwanted memories in post-traumatic-stress-disorder.

## Figures and Tables

**Figure 1 cells-10-02299-f001:**
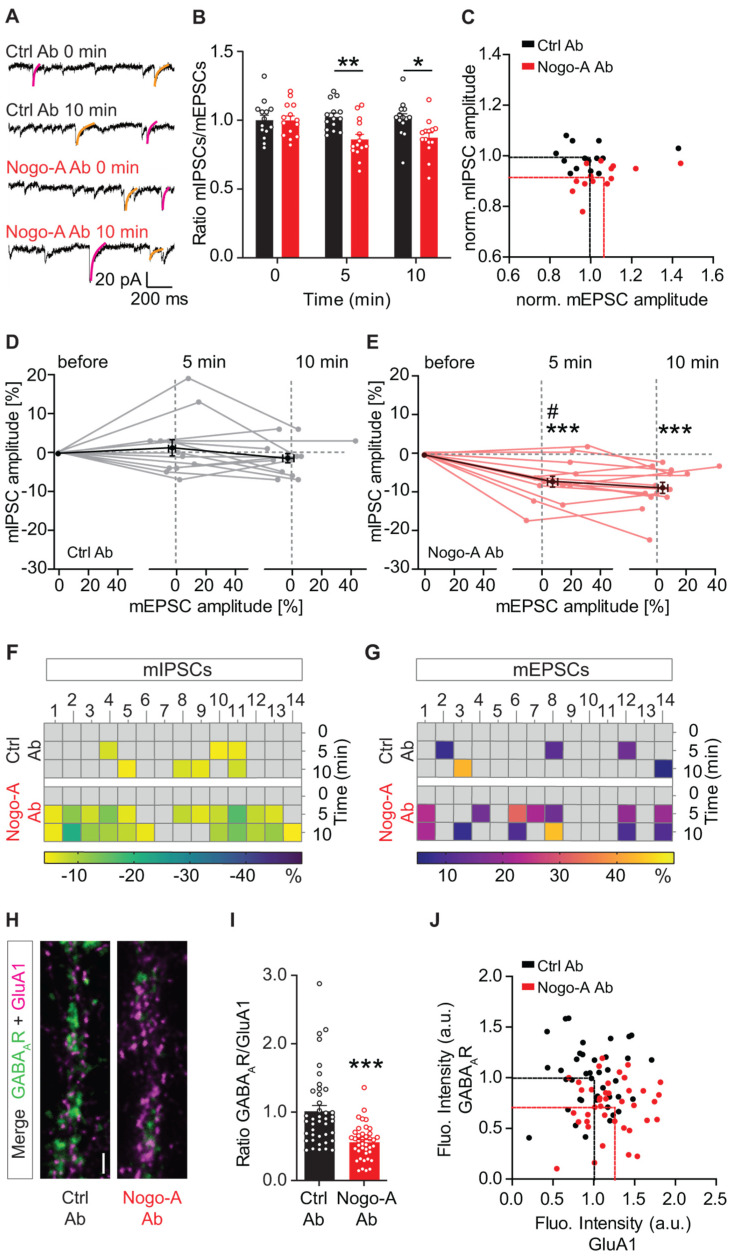
Nogo-A loss-of-function shifts the excitation/inhibition balance toward a higher excitation at a single cell level. (**A**–**C**) Patch-Clamp recordings in organotypic hippocampal cultures of simultaneously measuring mEPSCs and mIPSCs before (0 min) and after (5–10 min) control antibody, in all experiments mouse IgG1, FG12 (black, *n* = 14) or Nogo-A mouse IgG1, 11C7 function-blocking antibody (red, *n* = 14) treatment. (**A**) Example traces of mPSCs recorded by whole-cell patch clamp. Each trace shows representative mEPSCs (magenta) and mIPSCs (orange) distinguished by their decay time according to fast (mEPSCs < 25 ms) and slow (mIPSC > 25 ms) decay kinetics. Scale bars are 20 pA vertical and 200 ms horizontal (**B**) Normalized data for mIPSC and mEPSC ratio. (**C**) Normalized values for mIPSC and mEPSC amplitude plotted individually for each recorded neuron and the mean values visualized by the dotted line. (**D**,**E**) The percent change in the amplitude of mEPSC and mIPSC of individual neurons over time and their respective mean values ((**D**) Ctrl-Ab: black, (**E**) Nogo-A Ab: dark red, mEPSCs: # *p*, mIPSCs: *** *p*). (**F**,**G**) The percent change of individual recorded neurons in mIPSCs (**F**) and mEPSCs (**G**) exceeding average ± SEM color coded by heat mapping changes between 0–100% changes. (**H**–**J**) Live-cell labeling of surface GABA_A_ and GluA1 receptors in dissociated hippocampal neurons after 10 min Nogo-A loss-of-function (red, *n* = 40) compared to the control condition (black, *n* = 40). (**H**) Representative image of surface stained GABA_A_Rs (green) and AMPARs subunit GluA1 (magenta). All of the images underwent deconvolution and were equally increased in brightness and contrast by the same absolute values. Scale bars, 2 µm. (**I**) Normalized values for fluorescence intensity of GABA_A_R cluster and GluA1 immuno-positive puncta ratio upon Nogo-A loss-of-function for 10 min. (**J**) Normalized fluorescence intensity of GABA_A_R and GluA1 positive clusters plotted individually for each recorded neuron and the mean alteration visualized by the dotted line. Values represent means ± SEMs. # *p* < 0.05, * *p* < 0.05, ** *p* < 0.01, *** *p* < 0.001; *n* = number of neurons.

**Figure 2 cells-10-02299-f002:**
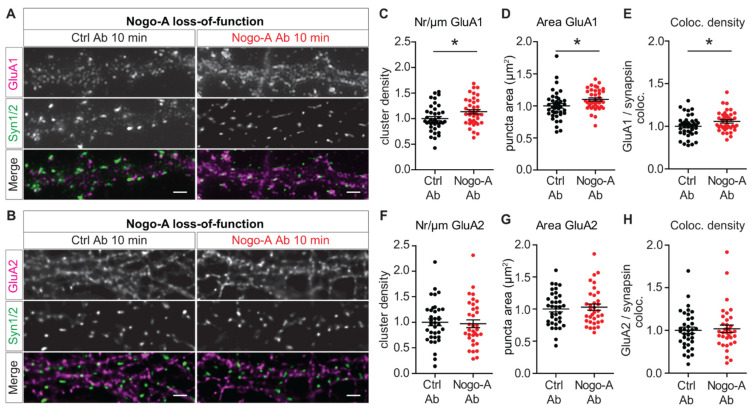
Nogo-A regulates the synaptic insertion of calcium permeable-AMPARs. (**A**,**B**) Live-cell immunolabeling of surface AMPAR subunit (magenta) GluA1 (**A**) or GluA2 (**B**) followed by immunofluorescence for presynaptic marker synapsin (Syn1/2; green) and their merged images (bottom) in primary hippocampal neurons treated for 10 min either with the control (left) or the Nogo-A function-blocking (right) antibody. For illustration, all images underwent deconvolution and were equally increased in brightness and contrast by the same absolute values for visibility. Scale bar 2 μm. (**C**,**D**) Normalized data for density (**C**) and fluorescence intensity (**D**) of GluA1 immuno-positive puncta in hippocampal neurons treated with either control antibody (black, *n* = 40) or Nogo-A function-blocking antibody (red, *n* = 39) for 10 min. (**E**) Normalized values for the density of GluA1 clusters colocalized with Syn 1/2 immuno-positive puncta. (**F**,**G**) Normalized GluA2 cluster density (**F**) and fluorescence intensity (**G**) in hippocampal neurons upon 10 min application with control antibody (black, *n* = 36) or Nogo-A function-blocking antibody (red, *n* = 35). (**H**) Normalized density of GluA2 immuno-positive puncta colocalized with Syn 1/2. Data are presented as mean ± SEM. * *p* < 0.05.

**Figure 3 cells-10-02299-f003:**
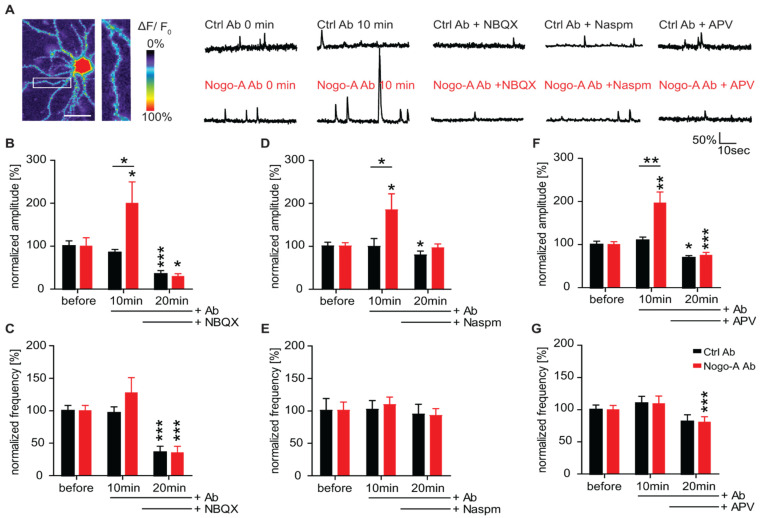
Nogo-A neutralization results in a calcium permeable-AMPAR dependent increase in Ca^2+^ dynamics. (**A**) Representative pseudo-colored image for Ca^2+^ response of a GCaMP5g transfected neuron. Scale bar 10 µm. Zoomed in region of the cell (white square) shows fluorescence intensity change of dendritic spines according to the heat map. The fluorescence intensity change (ΔF/F_0_) is displayed for 0–100%. Ca^2+^-imaging traces upon Nogo-A loss-of-function (red, bottom) or the control condition (black, above) both before (0 min) and after the antibody treatment (10 min) and the additional blocking of AMPARs (NBQX), CP-AMPARs (Naspm) or NMDARs (APV). The percentage change in fluorescence intensity over time is illustrated. Scale bars are 50% vertical and 10 sec horizontal. (**B**–**G**) Normalized change in Ca^2+^ transient amplitude and frequency upon either control (black) or Nogo-A function-blocking (red) antibody. Antibody treatments were performed for a total time of 20 min with additional wash-in of NBQX (B, C; Ctrl Ab: *n* = 11; Nogo-A Ab: *n* = 11), Naspm (D, E; Ctrl Ab: *n* = 15; Nogo-A Ab: *n* = 16) and APV (F, G; Ctrl Ab: *n* = 10; Nogo-A Ab: *n* = 16) for 10 min. Data are presented as mean ± SEM. * *p* < 0.05, ** *p* < 0.01, *** *p* < 0.001; *n* = number of neurons.

**Figure 4 cells-10-02299-f004:**
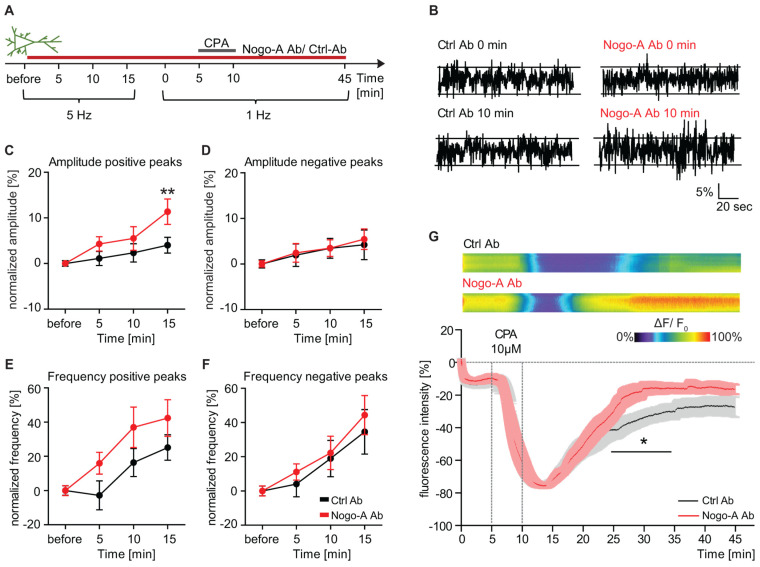
Ca^2+^ influx *versus* intracellular Ca^2+^ release upon Nogo-A loss-of-function. (**A**) Time scheme of live-cell imaging of a transfected neuron with the genetically encoded ER-targeted Ca^2+^ sensor ER-GCaMP6-150 in hippocampal primary cell culture. After baseline recording, the neuron was recorded at 5 Hz after control or Nogo-A antibody application for 15 min. As second part of the experimental setup, the same neuron was recorded at 1 Hz for 45 min and the additional treatment with Cyclopiazonic acid (CPA; 10 µM) for a total of 5 min. (**B**) Example traces for changes in relative fluorescence intensity of ER-mediated Ca^2+^ transients before and after 10 min treatment with the control (black) or the function-blocking antibody Nogo-A (red). The background noise is delineated by the dashed lines and serves as a threshold for the peaks that exceed the defined range and are counted for their frequency and measured amplitude of Ca^2+^ transients. Scale bar ΔF/F0 5% vertical and 20 sec horizontal. (**C**–**F**) Normalized values for frequency and amplitude of ER-mediated Ca^2+^ transients upon control antibody (black, *n* = 10) or Nogo-A function-blocking antibody (red, *n* = 10) over time. Percentage change in (**C**) amplitude of positive directed peaks, (**D**) amplitude of negative directed peaks, (**E**) frequency of positive directed peaks and (**F**) frequency of negative directed peaks. (**G**) Representative pseudo line-scans of ER-GCaMP6-150 transfected neuron representing changes in Ca^2+^ fluorescence of control antibody or Nogo-A function-blocking antibody upon CPA wash-in and wash-out. Heat map for a range of 0–100% intensity change is displayed. Averaged percent change of fluorescence intensity tracked over a time course of 45 min upon treatment with either Ctrl (grey, *n* = 9) or Nogo-A function-blocking antibody (red, *n* = 9). All data are presented as mean ± SEM. * *p* < 0.05.

**Figure 5 cells-10-02299-f005:**
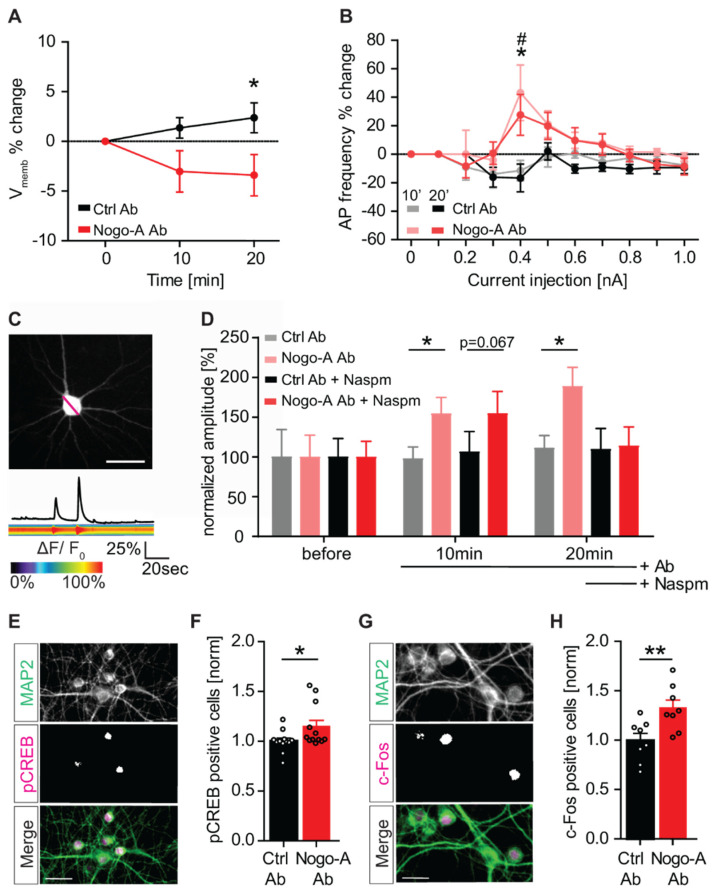
Nogo-A loss-of-function increases neuronal excitability and neuronal activation. (**A**,**B**) Patch-Clamp electrophysiology recordings of action potential (AP) threshold and frequency upon treatment with either control (black) or Nogo-A function-blocking antibody (red). (**A**) The normalized data show the percentage change of the action potential threshold (V_memb_) between both antibody conditions (*n* = 11) up to 30 min of treatment. (**B**) The frequency of APs was determined by applying current injections in 0.1 nA steps after 10 min control (grey) and Nogo-A (light red) antibody application, as well as after 20 min (Ctrl-Ab: black, *n* = 11; Nogo-A Ab: red, *n* = 10) 10 min: * *p*, 20min: # *p*. (**C**,**D**) Measurement of normalized Ca^2+^ transient amplitude percent change in primary hippocampal neurons transfected with GCaMP5g treated either with Nogo-A (red) or control (black) antibody. Antibody treatments were performed for a total time of 40 (**C**) or 20 min with an additional wash in of the CP-AMPAR blocker Naspm after 10 min (**D**). (**E**,**G**) Immunocytochemical staining for MAP2 (green) and pCREB (**E**) or c-Fos (**G**) (magenta) and their merged image (bottom). For illustration, all images underwent deconvolution and were increased in brightness and contrast for visibility. Scale bar 10 µm. (**F**,**H**) Normalized data for the number of pCREB (**F**) of c-Fos (**H**) positive neurons, after 10 min treatment with either Nogo-A-blocking (red) or control antibody (black). Each dot represents one coverslip. Data are presented as mean ± SEM. * *p* < 0.05, ** *p* < 0.01, # *p* < 0.05.

## Data Availability

All data supporting the described results are included in the manuscript. This study did not generate any new unique reagent.

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
