# Peer review of "Nogo-A Modulates the Synaptic Excitation of Hippocampal Neurons in a Ca2+-Dependent Manner"

_cells, 2021, doi:10.3390/cells10092299_

Round 1

Reviewer 1 Report

In the study titled “Nogo-A modulates the excitability of hippocampal neurons in a Ca2+-dependent manner”, Metzdorf K  et al have explored the molecular mechanism underlying the Nogo-A-regulated excitation/inhibition balance during plasticity in a model of pyramidal hippocampal neurons.

The previous studies have demonstrated the role of Nogo-A-promoted restriction of synaptic plasticity.

Their novel results showed that 1) Nogo-A loss-of-function leads to higher excitation following a decrease in GABAARs at inhibitory and an increase in the GluA1, but not GluA2 AMPAR subunit at excitatory synapses; 2) blocking Nogo-A in pyramidal hippocampal neurons increases neuronal excitability at the single-cell level; 3) release of Ca2+ from internal stores is not involved downstream of Nogo-A; 4) the effect of Nogo-A in regulating inhibitory synaptic strength and neuronal activation depends upon the influx of extracellular Ca2+, via a shift in the subunit composition toward a higher presence of Ca2+-permeable AMPARs at synapses; 5) Increased neuronal activation upon Nogo-A loss-of-function prompts the phosphorylation of the transcription factor CREB and the expression of c-Fos.

 Their main conclusion is that Nogo-A signaling stabilizes the Excitatory/ Inhibitory balance by controlling the neurotransmitter receptor dynamics in a Ca2+-dependent manner. Moreover, in the context of the implementation of unwanted memories in posttraumatic-stress-disorder, Nogo-A-modulated neuronal excitability, as well as pCREB and c-FOS levels results are extremely interesting.

Minor comments: The authors should add information on used anti-Nogo-A and anti-synapsin.

Author Response

Please see the attachment for reply to the review report

Reviever comment: The authors should add information on used anti-Nogo-A and anti-Synapsin.

Reply: We thank the reviewer for his positive feedback and hope that we can respond satisfactorily to his comment on the antibodies.

A paragraph is included in the material and method section describing the function-blocking antibody against Nogo-A.

For your information we copied the paragraph here below:

“METHOD DETAILS

Antibody and peptide treatment

Nogo-A loss-of-function was achieved via the application of the Nogo-A-NiG specific, monoclonal function-blocking antibody 11C7. The 11C7 antibody is raised against a 18-aa peptide located in the inhibitory domain at the N-terminus of Nogo-A (11C7, mouse IgG1; [72,75,76]; gift from Novartis Pharma AG, Basel, Switzerland to Martin Schwab). A mouse IgG1 against anti-BrdU (FG12, gift from Novartis Pharma AG, Basel, Switzerland to Martin Schwab) was used as control for the Nogo-A loss-of-function experiments.”

The antibody used in this study to label the presynaptic marker Synapsin 1/2 is also described in the Material and Method part: “anti-synapsin 1/2 antibody (Chicken anti-Synapsin1/2, Synaptic Systems, Cat# 106006, 1:1000)”.

We hope that these descriptions are sufficient.  However, please let us know if we should give a more detailed information.

Reviewer 2 Report

This is an interesting contribution showing that acute down-regulation of Nogo-A exerts significant effects on the balance between excitation / inhibition by specifically modulating the presence of Ca2+ permeable / Ca2+ impermeable glutamate receptors.

One general concern is that the degree of effective Nogo-A downregulation is not known and not controlled. This may explain the sometimes rather small size of effects. This should be discussed in detail as a major limitation.

Otherwise, I have only a few remarks.

Please provide a brief description in the abstract of what kind of protein Nogo-A is.

In the abstract, please describe briefly the system worked on (i.e. how Nogo-A was reduced, mice, slices, …)

Line 35: Ref 4 does not appear to be on disease, rather ref 3 seems so.

I think the references should only have the initials of first names.

Lines 50-54, and throughout the manuscript: please state which animals these studies are referring to.

Check the use of the comma. Normally there is no comma after “that”.

Line 62: it’s “associated with“ (not “to”).

Line 109: the concentration of 1 mM of these inhibitors seems extremely high.

Fig. 1B. Please show all data points.

Fig. 1: what are the “ns” referring to? Number of different cells?

Author Response

Please see see attachment for Reply to the Review Report

Reviewer comments:

  1. One general concern is that the degree of effective Nogo-A downregulation is not known and not controlled. This may explain the sometimes rather small size of effects. This should be discussed in detail as a major limitation.

Reply: We would like to apologize for not writing it clearly enough and for generating the confusion. Indeed, in our study Nogo-A is not downregulated. Rather, in all the experiments presented in our manuscript Nogo-A is blocked by application of a function-blocking antibody. We have now specified this again in the abstract for clarity.

Regarding the effect size in our results: the effect of blocking Nogo-A on the localization of AMPAR and GABAAR may appear to be rather small and smaller than the effects seen on the size of the mEPSCs and mIPSCs measured by Patch-Clamp. We think that this might be due to the different culture techniques used. While the immunostainings of AMPAR and GABAARs were done in primary hippocampal cultures, the Whole-cell patch clamp were done in organotypic culture slices, where the different subregions of the hippocampus are preserved. Zagrebelsky and colleagues (Journal of Neuroscience, 2010) demonstrated application of the Nogo-A functional antibody 11C7 to organotypic slice cultures leads to in alterations in dendritic structure, spine type distribution and axon morphology which were stronger for pyramidal neurons in the CA3 than in the CA1 region. Moreover, a similar effect size upon Nogo-A loss-of-function by the application of blocking antibodies was described also for other parameters such as the changes in dendritic spine density and length (Kellner et al., 2016). Indeed, Nogo-A is not the only ligand binding to the two receptors NgR1 (also MAG and OmGP) and S1PR2 (S1P) suggesting the possibility for compensatory processes which may justify the effect size.

  1. Please provide a brief description in the abstract of what kind of protein Nogo-A is.

 Reply: Following the reviewer comment to briefly describe Nogo-A, we have adapted the following sentence in the abstract: “In this context, the neurite growth inhibitor membrane protein Nogo-A modulates synaptic plasticity, strength, and neurotransmitter receptor dynamics.

  1. In the Abstract, please describe briefly the system worked on (i.e. how Nogo-A was reduced, mice, slices, …)

Reply: We have now changed the abstract by adding the information about the type of neuronal preparation used for each of the experiments performed. We hope that our more detailed description has increased the transparency of our experiments. Since we mainly worked in primary mouse hippocampal cultures and organotypic slice cultures, we explicitly mentioned the culture system in the results section in addition to the material and methods section for the corresponding experiments.

  1. Line 35: Ref 4 does not appear to be on disease, rather Ref 3 seems so.

Reply: We agree with the reviewer that Ref 3 Sohal et al, 2019 is also a useful reference regarding the mentioned neuropsychiatric disorders. Also discussed in Sohal et al. 2019 are specific developmental mechanisms that reduce inhibition. We find that the reference makes an important contribution to both statements and have added Sohal et al. 2019 in the context of pathophysiology, as E/I balance affects cortical and hippocampal functions.

Ref 4 Jacob et al. 2008 is a review highlighting, among other things, a number of neurological disorders, including epilepsy and schizophrenia, in which alterations in GABAAR trafficking occur.  We have now additionally labelled this reference as a review to avoid misunderstandings.

  1. I think the references should only have the initials of first names.

Reply: We would like to thank the reviewer who brought the citation style to our attention. We have adapted the references in the text as well as in the reference list according to the format guidelines.

  1. Lines 50-54, and throughout the manuscript: please state which animals these studies referring to.

Reply: Both in the introduction and in the discussion, we now made clear which animal system was used. We did this for our study and also for the references given. Especially in the discussion, we now make it clear right at the beginning that our study describes in vitro experiments of hippocampal mouse neurons.

  1. Check the use of the comma. Normally there is no comma after ”that”.

Reply: We would like to thank the reviewer for pointing this out and erased the comma after “that” throughout the entire manuscript.

  1. Line 62: it’s “associated with” (not “to”).

Reply: Thank you for mentioning it, we have changed it.

  1. Line 109: The concentration of 1mM of these inhibitors seems extremely high.

Reply: The reviewer is referring to the antimitotic drugs added to the organotypic cultures to reduce the over proliferation of glia cells and mentioned in the Material and Method part: Organotypic hippocampal slice culture: After 72 h, a mixture of antimitotic drugs (cytosine arabinoside, uridine, and fluorodeoxyuridine; 1 mM each) was applied to reduce the number of non-neuronal cells.

We are sorry for the confusion. 1mM refers to the concentration of each of the drugs in the stock solution. The drugs are added to the medium at a final concentration of 1µM. We have adjusted that in the manuscript accordingly: “After 72 h, a mixture of antimitotic drugs (cytosine arabinoside, uridine, and fluorodeoxyuridine; stock solution of 1 mM each) was applied at a final concentration of 1 µM each to reduce the number of non-neuronal cells.

See for reference:

Llano, I., Marry, A., Johnson, J.W., Ascher, P. and G~ihwiler, B. (1988) Patch-clamp recording of amino acid-activated responses in organotypic slice cultures. Proc. Natl. Acad. Sci. USA, 85, 3221-3225.

  1. 1B: Please show all data points.

Reply: We have adjusted the figure accordingly. Figure 1b and also 1I now show the individual data points in addition to the bar chart.

  1. Fig 1: what are the “ns” referring to? Number of different cells?

Reply: The “n” is referring to the number of neurons, that were either patched in the whole-cell patch clamp recordings or imaged after performing immunolabelling for the surface receptors. We now have added this information in the figure legend.